# Chemical Structures of Lignans and Neolignans Isolated from Lauraceae

**DOI:** 10.3390/molecules23123164

**Published:** 2018-11-30

**Authors:** Ya Li, Shuhan Xie, Jinchuan Ying, Wenjun Wei, Kun Gao

**Affiliations:** 1State Key Laboratory of Applied Organic Chemistry, College of Chemistry and Chemical Engineering, Lanzhou University, Lanzhou 730000, China; yingjch16@lzu.edu.cn (J.Y.); weiwj14@lzu.edu.cn (W.W.); 2Lanzhou University High School, Lanzhou 730000, China; xieshzb@163.com

**Keywords:** lignans, neolignans, Lauraceae, chemical components, chemical structures

## Abstract

Lauraceae is a good source of lignans and neolignans, which are the most chemotaxonomic characteristics of many species of the family. This review describes 270 naturally occurring lignans and neolignans isolated from Lauraceae.

## 1. Introduction

Lignans are widely distributed in the plant kingdom, and show diverse pharmacological properties and a great number of structural possibilities. The Lauraceae family, especially the genera of *Machilus*, *Ocotea*, and *Nectandra*, is a rich source of lignans and neolignans, and neolignans represent potential chemotaxonomic significance in the study of the Lauraceae. Lignans and neolignans are dimers of phenylpropane, and conventionally classified into three classes: lignans, neolignans, and oxyneolignans, based on the character of the C–C bond and oxygen bridge joining the two typical phenyl propane units that make up their general structures [1]. Usually, lignans show dimeric structures formed by a *β*,*β*’-linkage (8,8’-linkage) between two phenylpropanes units. Meanwhile, the two phenylpropanes units are connected through a carbon–carbon bond, except for the 8,8’-linkage, which gives rise to neolignans. Many dimers of phenylpropanes are joined together through two carbon–carbon bonds forming a ring, including an 8,8’-linkage and another carbon–carbon bond linkage; such dimers are classified as cyclolignans. When the two phenylpropanes units are linked through two carbon–carbon bonds, except for the 8,8’-linkage, this constitutes a cycloneolignan. Oxyneolignans also contain two phenylpropanes units which are joined together via an oxygen bridge. Herein, lignans and neolignans are classfied into five groups: lignans, cyclolignans, neolignans, cycloneolignans, and oxyneolignans on the basis of their carbon skeletons and cyclization patterns. The majority of lignans isolated from Lauraceae have shown only minor variations on well-known structures; for example, a different degree of oxidation in the side-chain and different substitutions in the aromatic moieties, including hydroxy, methoxy, and methylenedioxy groups. Since the nomenclature and numbering of the lignans and neolignans in the literature follow different rules, the trivial names or numbers of the compounds were used to represent them. Furthermore, the semi-systematic names of compounds and their corresponding names in the literature are summarized in the Appendix A. Herein, we give a comprehensive overview of the chemical structures of lignans and neolignans isolated from Lauraceae.

## 2. Lignans

This section covers lignans formed by an 8,8’-linkage between two phenyl propane units, which are subclassified according to the pattern of the oxygen rings as depicted in Figure 1. The semi-systematic names of those lignans without trivial names and their corresponding names in found in the literature are given in Appendix A.

### 2.1. Simple Lignans

Machilin A (**1**) was first obtained from the CHCl_3_-soluble portion of the methanolic extract of the bark of *Machilus thunbergii* collected at Izu Peninsula, together with *meso*-dihydroguaiaretic acid (**2**). The absolute structure of machilin A (**1**) was determined to be 2*S* and 3*R* (meso-form) [2]. Yu and Ma et al. also reported that the bark of *M. thunbergii* contained machilin A (**1**) and *meso*-dihydroguaiaretic acid (**2**). Furthermore, *meso*-dihydroguaiaretic acid (**2**) was found to have significant neuroprotective activity against glutamate-induced neurotoxicity in primary cultures of rat cortical cells and exerted diverse hepatoprotective activity, perhaps by serving as a potent antioxidant [3,4]. Activity-guided fractionation of the dichloromethane extract of the bark of *M. thunbergii* not only led to the isolation of machilin A (**1**) and *meso*-dihydroguaiaretic acid (**2**), but also *meso*-austrobailignan-6 (**3**) and *meso*-monomethyl dihydroguaiaretic acid (**4**). It was reported that *meso*-dihydroguaiaretic acid (**4**) showed potent inhibitory activity against DNA topoisomerase I and II in vitro at a concentration of 100 μM with inhibition ratios of 93.6 and 82.1%, respectively. Furthermore, *meso*-austrobailignan-6 (**3**) was referred to as *threo*-austrobailignan-6 (**10**) in the article [5]. Two diastereomeric dibenzylbutane lignans ((**5**) and (**6**)) which exhibited selective inhibition against COX-2 (cyclooxygenase) were obtained from the leaves of *Ocotea macrophylla* Kunth, which were collected in Nocaima county, Colombia [6]. Besides machilin A (**1**) and *meso*-dihydroguaiaretic acid (**2**), oleiferin C (**7**) also were found in the stem bark of *M. thunbergii* collected at Ulleung-Do, Kyungbook, Korea. Moreover, *meso*-dihydroguaiaretic acid (**2**) and oleiferin C (**7**) induced an apoptotic effect in HL-60 cells via caspase-3 activation [7]. *meso*-Dihydroguaiaretic acid (**2**), *threo*-dihyidroguaiaretic acid (**8**), sauriol B (**9**), and *threo*-austrobailignan-6 (**10**) were isolated from the ethanolic extract of the bark of *Nectandra turbacensis* (Kunth) Nees [8]. The leaves and root bark of *N. turbacensis* (Kunth) Nees collected in the city of Santa Marta (Magdalena, Colombia) contained *meso*-monomethyl dihydroguaiaretic acid (**4**), *threo*-dihyidroguaiaretic acid (**8**), austrobailignan-5 (**11**), and schineolignin B (**17**) [9]. Lignan **12** was first obtained from the leaves of *Apollonias barbujana* collected in San Andrésy Sauces [10]. Compounds **13**–**16** were found to occur in the trunk wood of *N. puberula.* Proof of the absolute structure of compound **13** relied on its acid catalyzed cyclization into (-)-galbulin, a tetralin-type neolignan of known absolute stereochemistry [11] (Figure 2). 

### 2.2. 7,7’-Epoxylignans

Nectandrin A (**19**) and nectandrin B (**20**) were first isolated from leaves and stems of *Nectandra rigida* Nees, along with galgravin (**18**) [12]. Nectandrin A (**19**) and nectandrin B (**20**), together with machilin F (**21**), machilin G (**22**), machilin H (**23**), and machilin I (**25**) were found to occur in the methanolic extract of the bark of *M. thunbergii* Sieb. et Zucc [13]. Galgravin (**18**), henricine (**26**), and veraguensin (**29**) were found in the leaves and root bark of *N. turbacensis* (Kunth) Nees [9]. Zuonin B (**24**), machilin F (**30**), and nectandrin B (**20**) were obtained from the stem bark of *M. thunbergii* [7]. Galgravin (**18**) and veraguensin (**29**), together with the 2,5-phenyl ring disubstituted lignans **27** and **28** were described for the first time from an ethanolic extract of the leaves *Ocotea foetens* [14]. Veraguensin (**29**) was first reported to be isolated from *Ocotea veraguensis* [15,16], and this compound was also found in *N. puberula* [11]. Verrucosin (**30**) was first gained from the benzene extract of branch wood of *Urbanodendron verrucosum*, together with austrobailignan-7 (**31**) and calopiptin (**32**). The structure of verrucosin (**30**) was established by comparison with the synthetic racemate and by the preparation of a dimethyl ether followed by a comparison of spectral data with published data to determine the absolute structure [17]. (+)-Galbacin (**33**), (+)-galbelgin (**34**), nectandrin A (**19**), nectandrin B (**20**), and machilin-G (**22**) were found to occur in the dichloromethane extract of the bark of *M. thunbergii* Sieb. et Zucc. Furthermore, nectandrin B (**20**) showed potent inhibitory activity against DNA topoisomerase I and II in vitro at a concentration of 100 μM, with inhibition ratios of 79.1 and 34.3%, respectively [3,4,5]. Beilschminol B (**35**) was first obtained from the roots of *Beilschmiedia tsangii* [18]. Odoratisol C (**36**), odoratisol D (**37**), and machilin-I (**25**) were obtained from the air-dried bark of the Vietnamese medicinal plant *M. odoratissima* Nees [19] (Figure 3). 

### 2.3. 7,9’-Epoxylignans

(-)-Parabenzoinol (**38**) was isolated from the fresh leaves of *Parabenzoin trilobum* Nakai, and its structure was elucidated by X-ray crystallographic analysis [20]. Actifolin (**39**) was identified in the stems of *Lindera obtusiloba*; moreover, its effect on tumor necrosis factor (TNF)-α and interleukin (IL)-6 as well as its inhibitory activity against histamine release were examined using human mast cells. Actifolin (**39**) suppressed the gene expressions of proinflammatory cytokines, TNF-α, and IL-6 in human mast cells [21,22] (Figure 4).

### 2.4. Lignan-9,9’-Olides

(-)-Parabenzlactone (**40**) and acetylparabenzylactone (**41**) were found in the fresh leaves of *P. trilobum* Nakai [23]. 5,6-Dihydroxymatairesinol (**42**) was found to occur in the methanolic extract of the stems of *L. obtusilob* [21] (Figure 4). 

### 2.5. 2.9,2’.9’-Diepoxylignans

The chromene dimer **43** was obtained from the ethanolic extract of the leaves of *Cinnamomu mparthenoxylon* (Jack) Meisn [24] (Figure 4).

### 2.6. 7.9’,7’.9-Diepoxylignans

The ethanol/H_2_O (9:1) extract of the fruits of *L. armeniaca* contained magnolin (**44**) and eudesmin (**45**) [25]. Phytochemical studies revealed the presence of sesamin (**46**) and *O*-methylpiperitol (**47**) in the ethanolic extract of the fruit of calyces of *N. amazonurn* [26]. Sesamin (**46**) was also found to occur in the CH_2_Cl_2_ extract of the bark of *M. thunbergii* Sieb. et Zucc [3,4]. Magnolin (**44**), eudesmin (**45**), sesamin (**46**), and *O*-methylpiperitol (**47**) were all found to occur in *Persea pyrifolia* Nees and Mart. ex Nees [27]. Phytochemical investigations of the methanolic extract of the leaves of *A. barbujana* resulted in the isolation of demethylpiperitol (**48**) [10]. The ethanolic extract of *Pleurothyrium cinereum* also contained (+)-demethylpiperitol (**48**), as well as (+)-de-4’’-O-methylmagnolin (**49**), which was found to be a potent COX-2/5-LOX dual inhibitor and platelet-activating factor (PAF)-antagonist (COX-2: IC_50_ = 2.27 μM; 5-LOX: IC_50_ = 5.05 μM; PAF: IC_50_ = 2.51 μM) (**49**) [6,28]. (+)-Syringaresinol (**50**) was isolated from the stems of *C. reticulatum* Hay [29]. (+)-De-4’’-O-methylmagnolin (**49**) and (+)-syringaresinol (**50**) both were found to occur in the methanolic extract of the stems of *Actinodaphne lancifolia* [30]. The leaves of *C. macrostemon* Hayata [31] and the stems of *C. burmanii* [32] both contained (+)-syringaresinol (**50**) and yangambin (**51**), which showed various pharmacological effects. Moreover, *C. burmanii* also contained (+)-sesamin (**46**) [32]. (+)-Demethoxyexcelsin (**52**), (+)-piperitol (**53**), and (+)-methoxypiperitol (**54**) were obtained from the bark and wood of *N. turbacensis*, together with (+)-sesamin (**46**) [33]. Epiyangambin (**55**), episesartemin (**56**), and yangambin (**51**) were isolated from the leaves of *O. duckei*, and yangambin (**51**) represented the major constituent [34]. Kwon et al. reported the isolation of (+)-syringaresinol (**50**) and pluviatilol (**57**) from the stems of *L. obtusilob*, and pluviatilol (**57**) showed cytotoxicity against a small panel of human tumor cell lines [21]. (+)-5-Demethoxyepiexcelsin (**58**) and (+)-epiexcelsin (**59**) were reported to be found in *Litsea verticillata* Hance, and (+)-5- demethoxyepiexcelsin (**58**) showed moderate anti-HIV activity with an IC_50_ value of 16.4 μg/mL (42.7 μM) [35]. (+)-Xanthoxyol (**60**), (+)-syringaresinol (**50**), and pluviatilol (**57**) were obtained from the stems of *L. obtusiloba* Blume. The effect of these compounds on tumor necrosis factor (TNF)-α and interleukin (IL)-6 as well as their inhibitory activity against histamine release were examined using human mast cells. Pluviatilol (**57**) inhibited the release of histamine from mast cells [22]. 4-Keto-pinoresinol (**61**) was isolated from the ethanolic extract of the leaves and twigs of *Litsea chinpingensis* [36] (Figure 5).

## 3. Cyclolignans 

There are three main types of cyclolignans isolated from nature, including 2,7’-cyclolignans, 2,2’-cyclolignans, and 7,7’-cyclolignans. Cyclolignans are not so common in Lauraceae. We have only retrieved less than 10 2,7’-cyclolignans isolated from Lauraceae. The semi-systematic names of those cyclolignans without trivial names and their corresponding names in the literature are given in Appendix A.

### 2,7.’-Cyclolignans

(-)-Isoguaiacin (**62**) and (+)-guaiacin (**63**) were isolated from the extract of the bark of *M. thunbergii* Sieb. et Zucc. These two compounds showed significant neuroprotective activities against glutamate-induced neurotoxicity in primary cultures of rat cortical cells [3,4]. (+)-Otobaphenol (**64**) and cyclolignans **65** and **66** were isolated from the ethanolic extract of *P. cinereum* [28]. Cinnamophilin A (**67**) was first reported to be obtained from the methanolic extract of roots of *Cinnamomum philippinense* (Merr.) Chang [37]. (-)-Aristoligone (**68**), (-)-aristotetralone (**69**), and (-)-cagayanone A (**70**) were obtained from the ethanolic extract of the leaves and twigs of *L. chinpingensis* [36] (Figure 6). 

## 4. Neolignans

Neolignans are widely distributed in the Lauraceae family, especially in the genera of *Aniba*, *Nectandra*, and *Ocotea.* The types of neolignans isolated from Lauraceae include 8,1’-neolignans, 8,3’-neolignans, 7,1’-neolignans, and 7,3’-neolignan (Figure 7). 3,3’-neolignans, which also exist in nature, have not been isolated from Lauraceae. The semi-systematic names of the abovementioned neolignans without trivial names and their corresponding names in the literature are given in Appendix A.

### 4.1. 8,1’-Neolignans

Burchellin (**71**) was first isolated from the trunk wood of *Aniba burchellii* Kosterm [38]. Burchellin (**71**) has also been found in the benzene extract of the trunk of an unclassified *Aniba* species collected in the vicinity of Manaus, Amazonas, along with compounds **72** and **73** [39,40]. Compounds **72** and **74** were obtained as a mixture from an unclassified Amazonian *Nectandra* species. As the analogous values for the mixture of compounds **72** and **74** were substantially identical to those of pure compound **72**, including the ORD (optical rotatory dispersion) curves, then the two compounds should have the same absolute configuration [41]. 3’-Methoxyburchellin (**75**) was first isolated from the stem bark of *O. veraguensis* [16]. Benzene extract of the trunk wood of *Aniba terminalis* [42] and ethanolic extract of the trunk wood of an *Aniba* species collected 130 km north of Manaus, Amazonas [43] both contained burchellin (**71**) and compound **76**. The trunk wood of *Ocotea catharinensis* yielded compound **77** [44,45]. Inspection of *Aniba simulans* revealed the occurrence of compounds **78**–**80** [46,47]. Armenin A (**81**) and armenin B (**82**) were first obtained from the benzene extract of the trunk wood of *Licaria armeniaca* [48]. The fruits of *L. armeniaca* yielded compounds **76** and **78** [25]. Compounds **74**, **78** and armenin C (**83**) were isolated from the fruits of *Aniba riparia* [49]. Compound **85** was isolated from the benzene extract of trunk wood of an unclassified *Aniba* species [50]. Canellin B (**86**) was first obtained from the benzene extract of the trunk wood of *Licaria canella* [51]. The trunk wood of an Amazonian *Aniba* species contained armenin A (**81**), armenin B (**82**), C (**83**), canellin B (**86**), canellin D (**87**), canellin E (**88**), porosin (**90**), and porosin B (**91**) [52]. Porosin (**90**) was first obtained from the wood of *Ocotea porosa* [53]. Porosin B (**91**) was first obtained from the branch wood of *U. verrucosum*, and porosin (**90**) also were found to exist in the same species [17]. The wood of *Ocotea catharinensis* yielded armenin B (**82**), canellin B (**86**), ferrearin C (**95**), ferrearin E (**96**), and compounds **92** and **93**. Moreover, the structures of compound **92** and ferrearin C (**95**) were certified by single-crystal X-ray analysis [45]. Ferrearin A (**99**) and ferrearin B (**100**) were first isolated from the trunk wood of the Amazonian *Aniba ferra* Kubitzki, together with compounds **85** and **92.** The relative structures of ferrearin A (**99**) and ferrearin B (**100**) were elucidated as structures of **97** and **98** [54], then revised as ferrearin A (**99**) and ferrearin B (**100**). Besides these two compounds, 3’-methoxyburchellin (**75**), compound (**77**), ferrearin C (**101**), and ferrearin D (**102**) were found to occur in the trunk wood of *Ocotea aciphylla* [55,56]. Burchelin (**71**), porosin (**90**), porosin B (**91**), and compounds **76**, **89**, **94**, **103**–**106**, **112**, and **113** all were identified in the trunk wood of *O. porosa*, collected from the Forest Reserve of the Botanical Institute, Sâo Paulo, Brazil [57,58]. Fifteen 8,1’-neolignans have been reported to be found in the bark and leaves of *O. porosa* harvested near Santa Maria, State of Rio Grande do Sul, Brazil, including burchellin (**71**), porosin (**90**), porosin B (**91**), and compounds **76**, **89**, **94**, and **105**–**113** [59] (Figure 8). 

### 4.2. 8,3’-Neolignans 

The benzene extract of trunk wood of *Licaria aritu* Ducke [60] and the EtOH/H_2_O (9:1) extract of the fruits of *N. glabrescens* contained licarin A (**114**) and licarin B (**115**) [26]. Licarin A (**114**) was also isolated from *N. rigida* Nees and is responsible for the major cytotoxic activity of crude extract of *N. rigida* Nees, displaying ED_50_ vs. KB cancer cell line at 7.0 μg/mL [12]. Machilin B (**116**) was obtained from the methanolic extract of the bark of *M. thunbergii* [2], as well as licarin A (**114**) and licarin B (**115**). Licarin A (**114**) showed significant neuroprotective activities against glutamate-induced neurotoxicity in primary cultures of rat cortical cells and induced an apoptotic effect in HL-60 cells via caspase-3 activation [4,7]. Licarin A (**114**) and licarin D (**117**) were found in branch wood of the shrub *U. verrucosum* [17]. Obovatifol (**118**), odoratisol-A (**119**), and (-)-licarin A (**114)** were obtained from the air-dried bark of the Vietnamese medicinal plant *Machilius odoratissima* Nees [19]. Besides licarin A (**114**) and licarin B (**115**), machilusol A (**120**), machilusol B (**122**), machilusol C (**123**), machilusol D (**124**), machilusol E (**125**), machilusol F (**126**), and acuminatin (**127**) were isolated from the stem wood of *Machilus obovatifolia*. Machilusols A–F showed moderate cytotoxic activity [61]. The dichloromethane extract of the bark of *M. thunbergii* Sieb. et Zucc also contained (-)-acuminatin (**127**), together with licarin A (**114**). (-)-Acuminatin exerted diverse hepatoprotective activities, perhaps by serving as a potent antioxidant [3,5]. Dihydrodehydrodiconifery alcohol (**129**) was found in the ethanolic extract of the leaves and twigs of *L. chinpingensis* [36]. Compound **130** was first obtained from the benzene extract of the trunk of an *Aniba* species collected in the vicinity of Manaus, Amazonas, along with acuminatin (**127**) and licarin D (**116**) [40]. *A. burchellii* Kosterm contains compounds **130** and **131**. The determination of their absolute stereochemistry relied on spectra and a preparation by thermolysis as well as the acid isomerization of burchellin (**71**) [62]. Denudatin B (**132**), as well as (+)-licarin A (**114)**, liliflol B (**121**), and (**+**)-acuminatin (**127**), have been found in leaves of *Nectandra amazonum* Nees [63]. Mirandin A (**133**) was proved to be the major neolignan of an unclassified *Nectandra* species, which grew at Rosa de Maio, a locality on the Manaus-Itacoatiara highway, Amazonas [41]. (+)-Mirandin A (**133**), (-)-licarin A (**114**), and (-)-licarin B (**115**) also were found to occur in the ethanolic extract of *P. cinereum* [28]. Licarin C (**128**), mirandin A (**133**), mirandin B (**134**), and compounds **135** and **146** were obtained from the benzene extract of *Nectundru mirunda* trunk wood [64]. Compounds **136** and **137** were found in the stem bark of *O. veraguensis* [16]. Furthermore, compound **136** also was found in the trunk wood of an *Aniba* species collected 130 km north of Manaus, Amazonas [43], and compound **137** was obtained from the wood of *O. catharinensis* [44] and the fruits of *O. veraguensis* [65]. Compounds **135**, **138, 139**, **141**, and **143**–**145** were isolated from the benzene extract of *Anibu simulans* trunk wood [46]. The extract of EtOH/H_2_O (9:1) of fruits of *L. armeniac* also provided compounds **136** and **138 [25]**. Obovaten (**142**), perseal D (**159**), perseal C (**160**), and obovatinal (**161**) were first obtained from the leaves of *Persea obovatifolia*; together with obovatifol (**118**), these compounds showed significant cytotoxicity against P-388, KB16, A549, and HT-29 cancer cell lines in vitro [66,67]. Compound **148** was isolated from the benzene extract of the trunk wood of *A. terminalis* [42]. Lancifolins A–F (**149**–**154**) were obtained from branches of the shrub *Aniba lancifolia* Kubitzki et Rodrigues [68]. Neolignan ketone **156** was found to exist in the chloroform extract of the bark of *Ocotea bullata* [69]. Ocophyllals A (**157**) and ocophyllals B (**158**), which have a C-1’ formyl side chain instead of a propenyl group, as well as (+)-licarin B (**115**) were observed to occur in the ethanolic extract from leaves of *O. macrophylla* [70]. Licarin A (**114**), licarin B (**115**), (7*R*,8*S*,1’*R*)-7,4’-epoxy-1’-methoxy-3,4-methylenedioxy-8,3’-neolign-8’-ene-6’(1’H)-one (**140)**, and compounds **130**, **147**, **155**, **162**, and perseal F (**163**) were obtained from *O. porosa* [57,58,59]. Compound **162** also was found to occur in the dichloromethane extract of the bark of *M. thunbergii* SIEB. et ZUCC [5]. Meanwhile, perseal F (**163**) and perseal G (**164**) were present in the chloroform-soluble portion of the stem wood of *M. obovatifolia* [71] (Figure 9). 

### 4.3. 7,1’-Neolignans

*Licaria chrysophylla* gave a considerable proportion of chrysophyllin A (**165**), which was the first type of 7,1’-neolignan to be obtained. Chrysophyllin B (**166**), chrysophyllon I-A (**167**), and chrysophyllon I-B (**168**) were also identified in *L. chrysophylla* [72,73]. The trunk wood of an Amazonian *Aniba* species collected in the vicinity of Manaus, Amazonas also contained chrysophyllin A (**165**) and chrysophyllin B (**166**) [52] (Figure 10). 

### 4.4. 7,3’-Neolignans

Chrysophyllon II-A (**169**) and chrysophyllon II-B (**170**) belonging to the category of 7,3’-neolignans, both were found to occur in the bark, wood, and fruit calyces of *L. chrysophylla* [73] (Figure 10). 

## 5. Cycloneolignans 

Cycloneolignans are responsible for the chemotaxonomic characteristics of some genera in the Lauraceae family, such as *Aniba, Licaria*, and *Nectandra*. Most cycloneolignans isolated from Lauraceae belong to the categories of 7.3’,8.1’-cycloneolignans and 7.3’,8.5’-cycloneolignans (Figure 11). Only two 7.1’,8.3’-cycloneolignans have been reported to be obtained from *O. bullata.* The semi-systematic names of those cycloneolignans without trivial names and their corresponding names in the literature are given in Appendix A.

### 5.1. 7.3’,8.1’-Cycloneolignans

Guianin (**171**) was first obtained from the wood of *Aniba guianensis* Aubl [74]. Meanwhile, 2’-epiguianin (**172)** was isolated from the leaves of *O. macrophylla* Kunth, which showed inhibition activity against the platelet-activating factor (PAF)-induced aggregation of rabbit platelets with an IC_50_ value of 1.6 μM [70]. Fourteen 7.3’,8.1’-cycloneolignans, including 3’-methoxyguianin (**173**) and compounds **174**–**176** and **178**–**187,** were isolated from the trunk wood of *O. porosa* collected from the mountainous Atlantic forest region of São Paulo State, where it is known as ‘canela parda’ [75,76]. Compound **192** was first obtained from the trunk wood of *O. porosa* collected from the Forest Reserve of Instituto BottInico (SHo Paulo, SP), along with guianin (**171**) and 2’-epiguianin (**172)** [58]. Compound **191** was first obtained from the benzene extract of *A. burchellii* Kosterm, together with guianin (**171**) [62]. Compounds **1****93**–**195** and **229** were found to be present in the benzene extract of the trunk of a unclassified *Aniba* species collected from the Ducke Forest Reserve, Manaus, Amazonas, as well as guianin (**171**) and 3’-methoxyguianin (**173**) [40]. Compounds **184**, **194**–**198**, and **202**–**207** were obtained from seed coat and dried fruit pulp of *O. veraguensis.* Since the author did not describe how to determine the absolute configuration of these compounds, their name should contain the addition of the prefix ‘rel’ [65]. Otherwise, compounds **188**, **191**, and **208**–**210** were isolated from the petrol and chloroform extract of the stem bark of *O. veraguensis* [16]. The trunk bark of *O. catharinensis* contained compounds **200**, **208**, **209**, as well as canellin-C (**212**) and 5-methoxycanellin-C (**213**), and the contents of all these compounds in the bark were over 0.01% [44,45]. Compounds **214**–**216**, **218**, and **228** were found to occur in the trunk wood of an *Aniba* species collected 130 km north of Manaus, Amazonas [43]. The benzene extract of trunk wood pertaining to an unclassified *Aniba* species collected from the Ducke Forest Reserve, Manaus yielded **217**, **219**, **221**, **222**, **224**, and methoxycanellin A (**226**) [50,77]. Compound **189** was first obtained from the ethanolic extract of wood of *Ocotea costulatum*, along with compound **221** [78]. The trunk wood of the Amazonian *A. ferra* Kubitzki contained *rel*-(7*S*,8*R*,1’*S*,2’*S*,3’*R*)-1’,2’-dihydro-2’-hydroxy-3,3’,5’- trimethoxy-4,5-methylenedioxy-7.3’,8.1’-cycloneolign-8’-ene- 4’(3’H)-one (**199**) and methoxycanellin A (**226**) [54]. Canellin A (**225**) and canellin C (**212**) were first obtained from the trunk wood of *L. canella* [51]. These two compounds have also been reported to be found in the trunk wood of *L. rigida* Kosterm. However, the relative structures of the compounds shown in the abovementioned article were different from those shown in other articles—the methyl occupied an exo-configuration and the aryl adopted an endo-configuration [79]. The trunk wood of the central Brazilian *O. aciphylla* also yielded canellin-A (**225**), as well as compound **208** and 3’-methoxyguianin (**173**) [55,56]. 2’-Epiguianin (**172)**, compounds **177**, **190**–**192**, **211**, **220**, **223**, **227**, and *rel*-(7*R*,8*R*,1’*S*,3’*S*)-5’-methoxy-3,4-methylenedioxy-7.3’,8.1’-cycloneolign-8’-ene-2’,4’(1’H,3’H)-dione (**229**) were obtained from the bark and leaves of *O. porosa* [59]. Compounds **194**, **201**, **230**, and **231** were found in the extract of EtOH/H_2_O (9:1) of the fruits of *L. armeniaca* [25]. The chloroform extract of the trunk wood of *L. armeniaca* yielded compound **232** [80] (Figure 12). 

### 5.2. 7.3’,8.5’-Cycloneolignans 

Macrophyllin B (**233**) was purified from an unclassified *Nectandra* species collected at Rosa de Maio, a locality on the Manaus-Itacoatiara highway (8 km), Amazonas [41]. Nectamazins A–C (**234**–**236**), macrophyllin B (**233**), denudanolide D (**237**), and kadsurenin C (**238**) isolated from leaves of *N. amazonum* Nees showed inhibition activity against the platelet-activating factor (PAF)-induced aggregation of rabbit platelets [63]. A phytochemical exploration of the leaves of *O. macrophylla* afforded ocophyllols A–C (**239**–**241**). Their absolute configurations were established by derivatizing them with (R)-and (S)-MTPA, and then analyzing the NMR data, as well as by a comparison of their circular dichroism (CD) data with that of a related compound whose absolute configuration was previously established by single-crystal X-ray analysis. Moreover, ocophyllols A–C (**239**–**241**) showed some inhibition activity against the platelet-activating factor (PAF)-induced aggregation of rabbit platelets [70]. Cinerin B (**242**), cinerin C (**243**), cinerin A (**244**), and cinerin D (**245**) were isolated from the leaves of *P. cinereum.* Again, their CD data was used to determine the absolute configuration of these compounds. Cinerin C (**243**) was the first known macrophyllin-type cycloneolignan, which was isolated from the trunk wood of *Licaria macrophylla* Kosterm and named as macrophyllin A [81]. Cinerin A–D also showed some inhibition activity against the platelet-activating factor (PAF)-induced aggregation of rabbit platelets [82]. Compound **246** and macrophyllin B (**233**) were identified in the ethanolic extract of leaves of *P. cinereum* [28] (Figure 13). 

### 5.3. 7.1’,8.3’-Cycloneolignans 

Ocobullenone (**247**) was the first naturally occurring bicyclooctanoid found to exhibit the 7.1’, 8.3’ linkage, and it was isolated from the chloroform extract of the bark of *O. bullata* [83]. Iso-ocobullenone (**248**) was also isolated from the chloroform extract of the bark of *O. bullata*, and its structure was confirmed by single-crystal X-ray analysis [69] (Figure 14). 

## 6. Oxyneolignans

An ether oxygen atom provides the linkage between the two phenylpropane units, giving rise to oxyneolignans. Oxyneolignans are rarely distributed in Lauraceae. Less than 10 oxyneolignans have been found to occur in the Lauraceae family, belonging to two categories: 7.3’,8.4’-dioxyneolignans and 8,4’-oxyneolignans (Figure 15). 

### 6.1. 7.3’,8.4’-Dioxyneolignans

The trunk wood of *L. rigida* Kosterm contained eusiderin (**249**) and eusiderin B (**250**) [79]. The trunk wood of an unclassified *Aniba* species collected at the Ducke Forest Reserve, Manaus also yielded the benzodioxane-type neolignan eusiderin (**249**), eusiderin-F (**251**), and eusiderin-G (**252**) [50,77]. Eusiderin (**249**) was also found to be present in the ethanolic extract of wood of *O. costulatum* [78] (Figure 16). 

### 6.2. 8,4’-Oxyneolignans

Machilin C (**253**), D (**254**), and E (**255**) were first obtained from the methanolic extract of the bark of *M. thunbergii* [2]. Odoratisol B was obtained from the air-dried bark of the Vietnamese medicinal plant *M. odoratissima* Nees. This compound showed the same relative structure as machilin C (**253**), but was termed odoratisol B in the article [19]. Perseal A (**256**) and perseal B (**257**), which have a C-1’ formyl side chain instead of a propenyl group, were isolated from the chloroform-soluble fraction of the leaves of *P. obovatifolia.* They showed significant cytotoxicity against P-388, KB 16, A549, and HT-29 cancer cell lines [67] (Figure 17). 

## 7. Uncommon Lignans

This section covers lignans and neolignans that contain uncommon skeletons. The molecular backbone of compounds **258**–**267** consists of a unique C6–C3 unit, and an ether oxygen atom provides the linkage between the phenyl and propyl groups. The semi-systematic names of those uncommon lignans without trivial names and their corresponding names in the literature are given in Appendix A.

Compounds **258**–**261** were isolated from the benzene extract of *A. simulans* trunk wood [46]. Compounds **259** and **260** were also found in the fruits of *L. armeniaca* [25] and the trunk wood of *N. mirunda*, respectively [64]. Compound **262** was isolated from the benzene extract of the trunk wood of *A. terminalis* [42]. Compounds **263** and **264** have been found in the trunk bark of *O. catharinensis* [45]. The stem bark of *O. veraguensis* also yielded compound **263 [16]**. Chrysophyllon III-A (**265**) and chrysophyllon III-B (**266**) have been found in trunk wood, bark, and fruit calyces of *L. chrysophylla* [73]. Compound **267** was obtained from the bark and leaves of *O. porosa* [59]. (+)-9’-O-trans-feruloyl-5,5’- dimethoxylariciresinol (**268**), which showed cytotoxicity against a small panel of human tumor cell lines with ED_50_ values around 10 μg/mL, was isolated from the stems of *L. obtusiloba* Blume [21,22]. Turbacenlignan A (**269**), a 7,8-secolignan, was isolated from the leaves and root bark of *N. turbacensis* (Kunth) Nees [9]. Cinnaburmanin A (**270**) was isolated from the roots of *C. burmanii* [84] (Figure 18).

## 8. Conclusions

A renewed interest in compounds isolated from natural resources has led to an enormous class of pharmacologically active compounds. Lignans and neolignans have been revealed to show significant pharmacological activities, including antitumor, anti-inflammatory, immunosuppression, cardiovascular, antioxidant, and antiviral activities [85,86]. The Lauraceae family, especially the genera of *Machilus*, *Ocotea*, and *Nectandra*, represents a rich source of lignans and neolignans. Moreover, neolignans are responsible for the potential chemotaxonomic significance found in the study of Lauraceae. Studies on lignans and neolignans in Lauraceae were mainly carried out in the 1980s. There have been more studies concerning the identification of lignans and neolignans in Lauraceae, but less on the biological activities of these compounds. Among the lignans and neolignans isolated from Lauraceae, the biological activities of sesamin and yangambin have been studied more, while there are relatively few articles published on other compounds. Sesamin, a 7.9’,7’.9-diepoxylignan present in many species in the Lauraceae family such as *N. amazonurn*, *M. thunbergii*, *P. pyrifolia*, *C. burmanii*, and *N. turbacensis*, showed significant anticancer properties [87]. Yangambin (**51**) which was the major constituent of *O. duckei*, showed diverse biological activities [88]. Therefore, it is extremely urgent to expand the scope of research on the lignans and neolignans in Lauraceae, with the aim of discovering all biological activities of these compounds. 

## Figures and Tables

**Figure 1 molecules-23-03164-f001:**
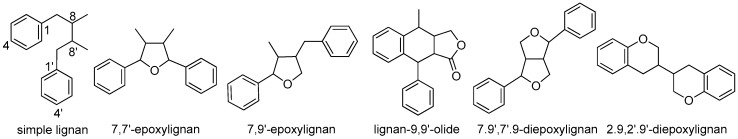
Subtypes of classical lignans.

**Figure 2 molecules-23-03164-f002:**
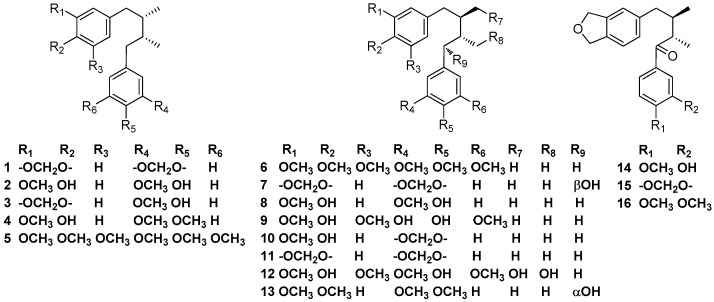
Chemical structures of simple lignans.

**Figure 3 molecules-23-03164-f003:**
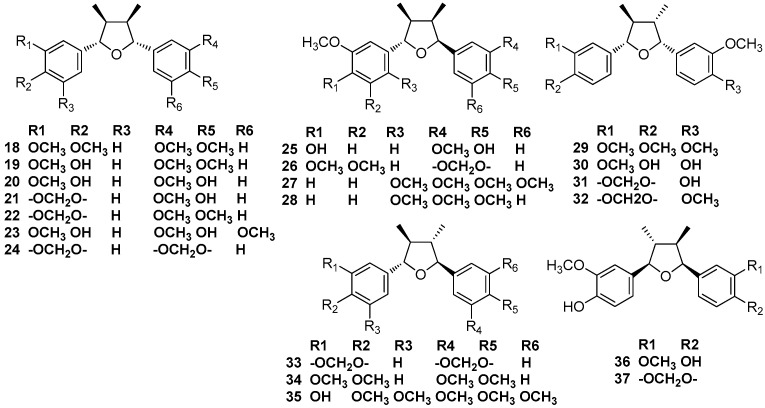
Chemical structures of 7,7’-epoxylignans.

**Figure 4 molecules-23-03164-f004:**
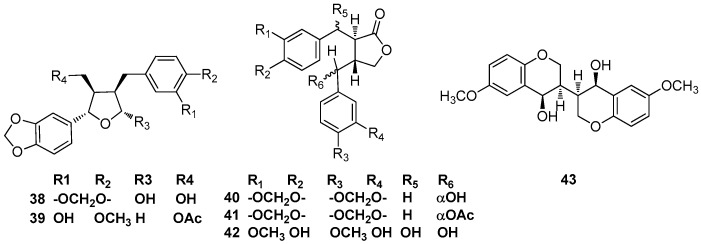
Chemical structures of 7,9’-epoxylignans, lignan-9,9’-olides, and 2.9,2’.9’-diepoxylignans.

**Figure 5 molecules-23-03164-f005:**
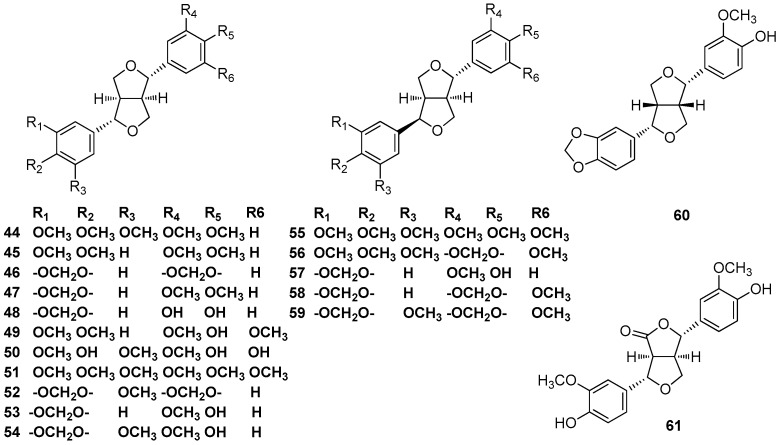
Chemical structures of 7.9’,7’.9-diepoxylignans.

**Figure 6 molecules-23-03164-f006:**
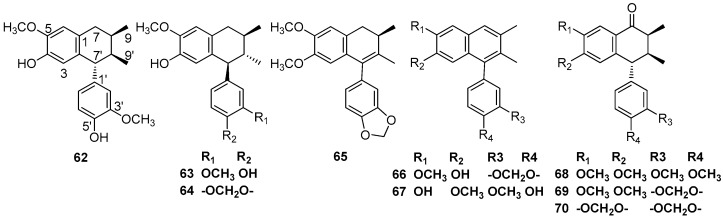
Chemical structures of 2,7’-cyclolignans.

**Figure 7 molecules-23-03164-f007:**
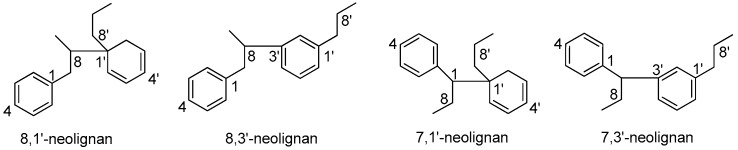
Subtypes of neolignans.

**Figure 8 molecules-23-03164-f008:**
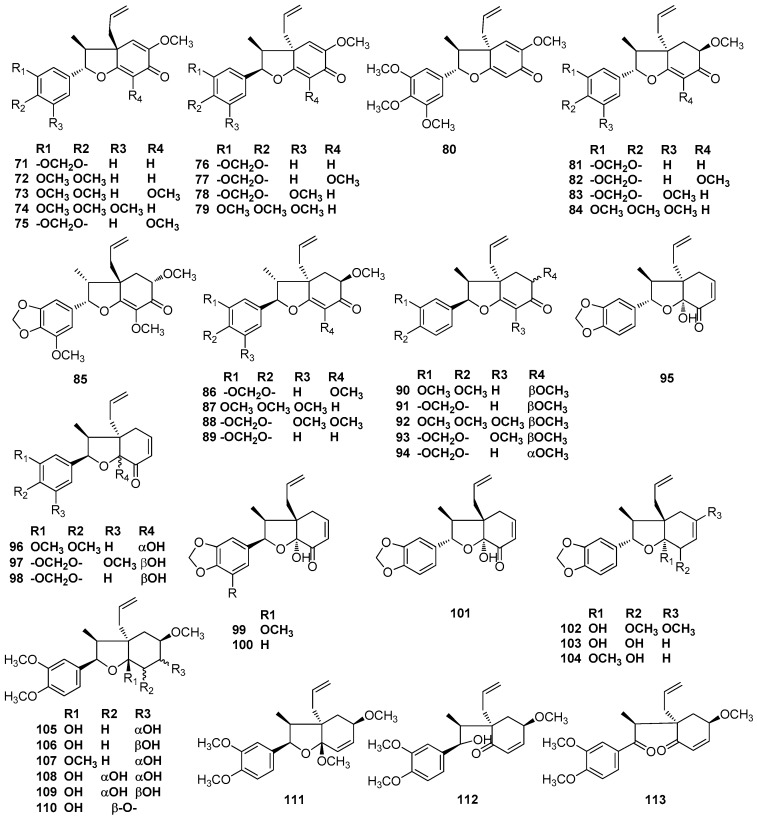
Chemical structures of 8,1’-neolignans.

**Figure 9 molecules-23-03164-f009:**
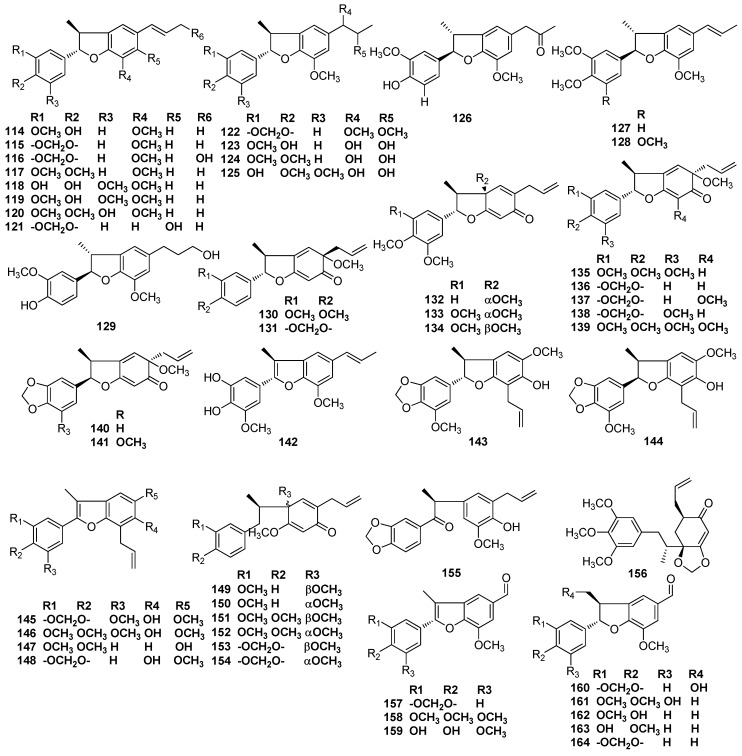
Chemical structures of 8,3’-neolignans.

**Figure 10 molecules-23-03164-f010:**
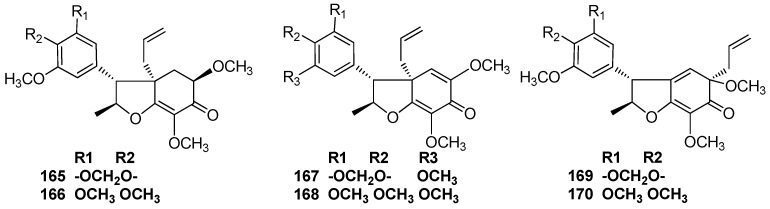
Chemical structures of 7,1’-neolignans.

**Figure 11 molecules-23-03164-f011:**
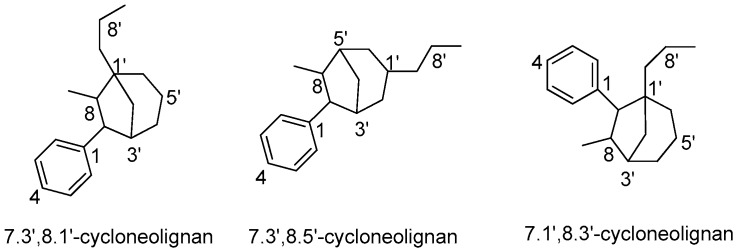
Subtypes of cycloneolignans.

**Figure 12 molecules-23-03164-f012:**
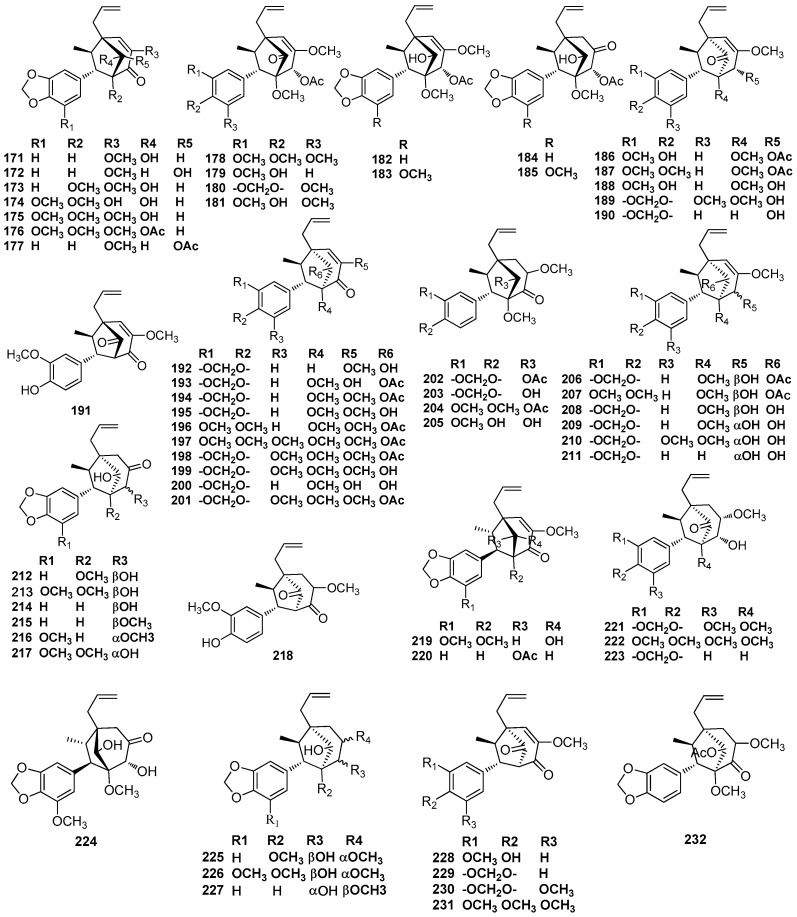
Chemical structures of 7.3’,8.1’-cycloneolignans.

**Figure 13 molecules-23-03164-f013:**
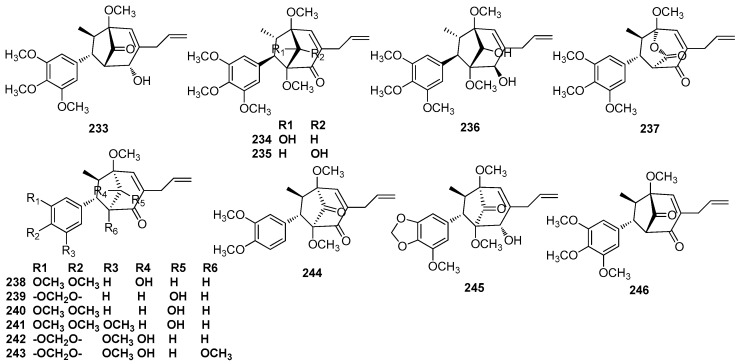
Chemical structures of 7.3’,8.5’-cycloneolignans.

**Figure 14 molecules-23-03164-f014:**
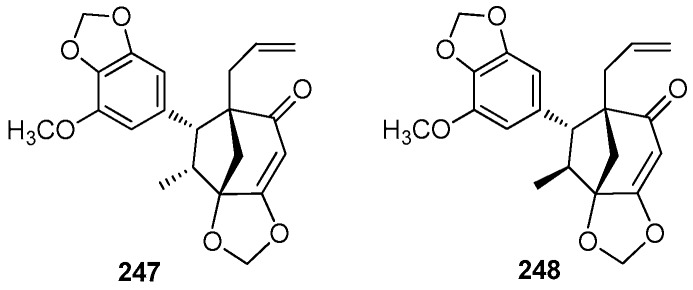
Chemical structures of 7.1’,8.3’-cycloneolignans.

**Figure 15 molecules-23-03164-f015:**
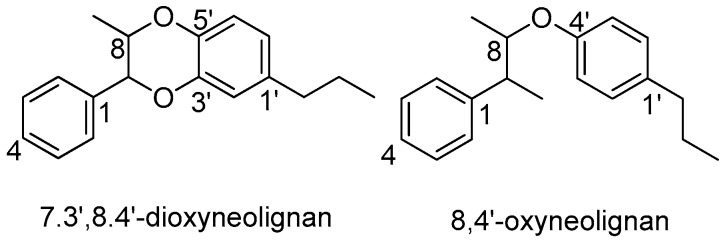
Subtypes of oxyneolignans.

**Figure 16 molecules-23-03164-f016:**
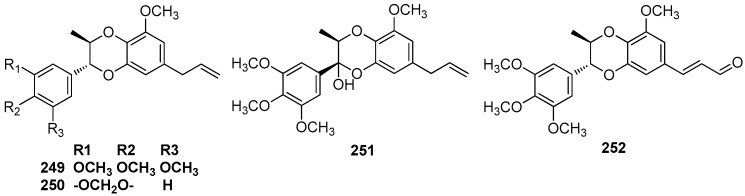
Chemical structures of 7.3’,8.4’-dioxyneolignans.

**Figure 17 molecules-23-03164-f017:**
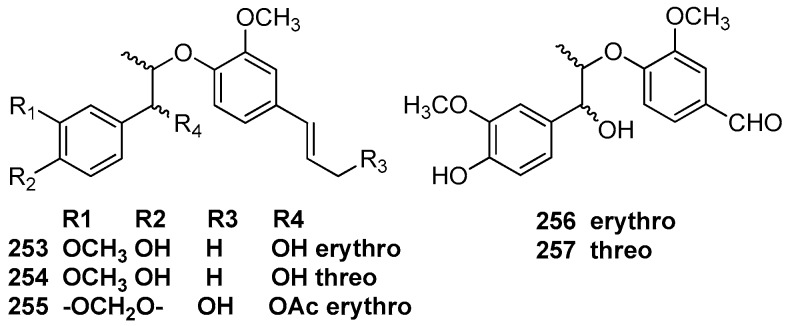
Chemical structures of 8,4’-oxyneolignans.

**Figure 18 molecules-23-03164-f018:**
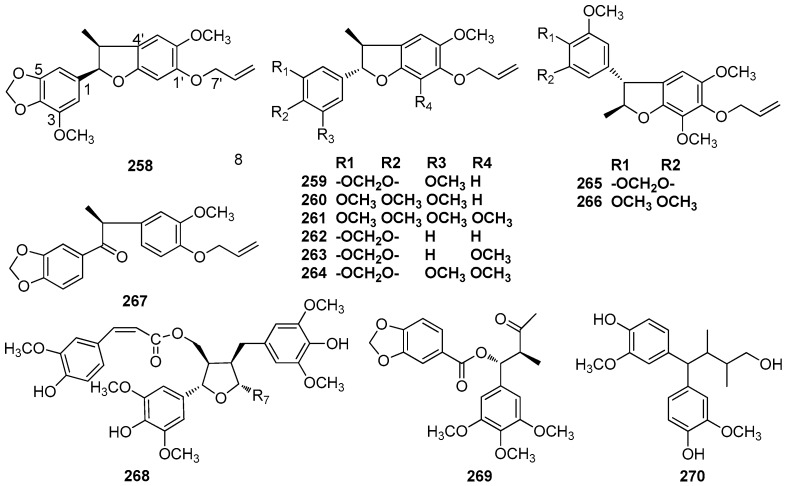
Chemical structures of uncommon neolignans.

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
