# Peer review of "Chemical Structures of Lignans and Neolignans Isolated from Lauraceae"

_molecules, 2018, doi:10.3390/molecules23123164_

Reviewer 1 Report

 The submitted manuscript:

"Ya Li, Shuhan Xie, Jinchuan Ying, Wenjun Wei, Kun Gao, Chemical Structures of Lignans and Neolignans Isolated from Lauraceae"

contains an overview of the Lauraceae species, the particular isolated lignans, methods of their isolation, their structures, and elucidations about the identification of the isolated compounds.

 Some remarks and comments

 * The overusing of the systematic chemical names. There are many systematic, chemical names of the particular, individual compounds in the manuscript. In my opinion, it very complicates the content and makes the text unreadable (especially fragments: 4.2, 5.1, 7). Besides the unreadability, this is also a potential source of mistakes.

The proposition would be as follows: only common names, structure drawings and numbers of the compounds should be used and discussed in the main text. To present the systematic, chemical names, the Supporting Information with them should be created.

 * The compounds in the manuscript are described in the following manner:

- the structural drawing/number/name,

- the taxonomic name of a plant, and a plant parts, the compound was isolated from,

- the isolation (extraction) details,

- the comments about the structure (esp. configuration) and the identification methods used.

The latter (structural problems) is not always present. In my opinion the comments about the method of the identification of a compound should be reported whenever possible.

 * The authors reported many references devoted to the presence of lignans in plants as well as methods of the lignan isolation. It is worthy to note that other review, italian article devoted to the lignans in brusera sp. has been recently published in Molecules: "Marcotullio, ..., An Ethnopharmacological, Phytochemical and Pharmacological Review on Lignans from Mexican Bursera spp., Molecules, 2018, 23, 1976".

It would be very advisible to add the above reference to this manuscript.

 * The phrase:

„Herein, we will give a comprehensive overiew of the chemical structures of lignans and neolignans isolated from Lauraceae.”

is doubled in lines 35-36 as well as 39-40.

 * The subtitle 3.1 is redundand because there are not subtitles 3.2, 3.3, et.ca.

 * line 75: acidatalysed --> acid catalysed

 Author Response

Dear reviewer

 Thanks for your critical comments and thoughtful suggestions. According to your comments, we have made careful modifications and revisions in the manuscript, and the major modifications made in the manuscript were typed in red. Below you will find our point-by-point responses to all comments and suggestions:

Some remarks and comments

  The overusing of the systematic chemical names. There are many systematic, chemical names of the particular, individual compounds in the manuscript. In my opinion, it very complicates the content and makes the text unreadable (especially fragments: 4.2, 5.1, 7). Besides the unreadability, this is also a potential source of mistakes.

The proposition would be as follows: only common names, structure drawings and numbers of the compounds should be used and discussed in the main text. To present the systematic, chemical names, the Supporting Information with them should be created.

Response: According to your kind suggestions, the common names or numbers of the compounds were used to represent them in the manuscript. And the semi-systematic names of those without trivial names and their corresponding names in literatures are summarized in tables in Supporting Information.

 * The compounds in the manuscript are described in the following manner:

- the structural drawing/number/name,

- the taxonomic name of a plant, and a plant parts, the compound was isolated from,

- the isolation (extraction) details,

- the comments about the structure (esp. configuration) and the identification methods used.

The latter (structural problems) is not always present. In my opinion the comments about the method of the identification of a compound should be reported whenever possible.

Response: Thanks for your valuable suggestions. The methods of the identification of the lignans and neolignans, especially the determination of the stereochemical configuration were mainly based on ORD and CD spectra, it's really not necessary to mention them moretimes. And relevant contents have been deleted.

* The authors reported many references devoted to the presence of lignans in plants as well as methods of the lignan isolation. It is worthy to note that other review, italian article devoted to the lignans in brusera sp. has been recently published in Molecules: "Marcotullio, ..., An Ethnopharmacological, Phytochemical and Pharmacological Review on Lignans from Mexican Bursera spp., Molecules, 2018, 23, 1976".

It would be very advisible to add the above reference to this manuscript.

Response: The reference has been added in the manuscript.

 * The phrase:

„Herein, we will give a comprehensive overiew of the chemical structures of lignans and neolignans isolated from Lauraceae.”

is doubled in lines 35-36 as well as 39-40.

Response: The phrase ‘Herein, we will give a comprehensive overiew of the chemical structures of lignans and neolignans isolated from Lauraceae.’ in lines 35-36 has been deleted.

* The subtitle 3.1 is redundand because there are not subtitles 3.2, 3.3, et.ca.

Response: The subtitle 3.1 has been deleted

 * line 75: acidatalysed --> acid catalysed

Response: ‘acidatalysed’ has been revised as acid catalysed

Thank you very much for your effort in handling our manuscript.

 Yours sincerely,

Ya Li

State Key Laboratory of Applied Organic Chemistry,

College of Chemistry and Chemical Engineering,

Lanzhou University, Lanzhou 730000, People's Republic of China.

 Reviewer 2 Report

Authors comprehensively described the chemical structures of lignans and neolignans from Lauraceae in the manuscript. They used the (semi-)systematic nomenclature for all compounds, which is too complex for reading. A table summarizing the systematic name vs chemical names will be helpful to readers.

Additionally, some discussion about biological activities of these compounds will increase the audience scope of this manuscript.

Author Response

Dear reviewer

 Thanks for your critical comments and thoughtful suggestions. According to your comments, we have made careful modifications and revisions in the manuscript, and the major modifications made in the manuscript were typed in red. Below you will find our point-by-point responses to all comments and suggestions:

Authors comprehensively described the chemical structures of lignans and neolignans from Lauraceae in the manuscript. They used the (semi-)systematic nomenclature for all compounds, which is too complex for reading. A table summarizing the systematic name vs chemical names will be helpful to readers.

Response: According to your kind suggestions, the trivial names or numbers of the compounds were used to represent them in the manuscript. And the semi-systematic names of those without trivial names and their corresponding names in literatures are summarized in tables in Supporting Information.

Additionally, some discussion about biological activities of these compounds will increase the audience scope of this manuscript.

Response: Studies on lignans and neolignans in Lauraceae were mainly carried out in the 1980s. And there were more studies on the identification of lignans and neolignans in Lauraceae, but less on the biological activities of these compounds. Among the lignans and neolignans isolated from Lauraceae, the biological activities of sesamin and yangambin have been studied more, while the other compounds are relatively few. The biological activities of them has been described in “conclusions”. And the biological activities of other compounds had been described in the manuscript.

 Thank you very much for your effort in handling our manuscript.

 Yours sincerely,

Ya Li

State Key Laboratory of Applied Organic Chemistry,

College of Chemistry and Chemical Engineering,

Lanzhou University, Lanzhou 730000, People's Republic of China.

Reviewer 3 Report

The present manuscript showed the chemical structures of lignans and neolignans isolated from Lauraceae cyclopaedically. It was important and acceptable for the journal. 

Author Response

Dear reviewer

        Thank you very much for your effort in handling our manuscript.

Yours sincerely,

Ya Li

State Key Laboratory of Applied Organic Chemistry,

College of Chemistry and Chemical Engineering,

Lanzhou University, Lanzhou 730000, People's Republic of China.

Round  2

Reviewer 1 Report

All, but one of the remarks, have been sufficiently considered and corrected.

About the identification of the lignans. The CD and ORD data are of course necessary, for the solution of chiral problems. But this is a review article with many source references. All described compounds have been someway and someday originally isolated from plants and identified by spectral methods. In my opinion, it would be interesting for readers to know how it was done.